# Probabilistic computing using $Cu_{0.1}Te_{0.9}$/$HfO_2$/Pt diffusive memristors

Kyung Seok Woo[1,2], Jaehyun Kim[1,2], Janguk Han[1], Woohyun Kim [1], Yoon Ho Jang[1] & Cheol Seong Hwang [1] ✉

A computing scheme that can solve complex tasks is necessary as the big data field proliferates. Probabilistic computing (p-computing) paves the way to efficiently handle problems based on stochastic units called probabilistic bits (p-bits). This study proposes p-computing based on the threshold switching (TS) behavior of a $Cu_{0.1}Te_{0.9}$/$HfO_2$/Pt (CTHP) diffusive memristor. The theoretical background of the p-computing resembling the Hopfield network structure is introduced to explain the p-computing system. P-bits are realized by the stochastic TS behavior of CTHP diffusive memristors, and they are connected to form the p-computing network. The memristor-based p-bit is likely to be '0' and '1', of which probability is controlled by an input voltage. The memristor-based p-computing enables all 16 Boolean logic operations in both forward and inverted operations, showing the possibility of expanding its uses for complex operations, such as full adder and factorization.

With the rapid development of big data, computing hardware that can handle complex tasks and exceed the conventional von Neumann architecture is being implemented[1,2]. The so-called memory wall issue is a critical problem for big-data-based computing in von Neumann computers. The Quantum computer shows the potential to exceed the performance of the classical computer[3–7]. However, maintaining the quantum-mechanically entangled state of the multiple quantum bits is challenging, and its cryogenic operating temperature leads to enormous energy consumption. Recently, probabilistic computing (p-computing) has been introduced to address the problems of the computing methods presented above[8,9]. The p-computing uses probabilistic bits (p-bits), which give '0' and '1' continuously changing over time. The p-bit has both the probability of being '0' and '1', and an input variable can change these probabilities. Its behavior is similar to that of a binary stochastic neuron in a neural network or neuromorphic computing system[10,11]. A magnetic tunnel junction (MTJ) was most recently used as the stochastic element in p-bits[12]. By reducing an energy barrier, which controls the resistance states of the MTJ, the magnetization direction of the MTJ fluctuated even with the thermal noise. The stochastic MTJ was connected with an n-type metal-oxide-semiconductor (NMOS) transistor to form a three-terminal p-bit. However, based on the p-bit principle, there is no reason for the MTJ to

be the only viable p-bit source. Any stochastic electronic device can be used if an external input voltage can control its internal state. Table 1 shows the comparison of different computation methods.

Memristor is a strong candidate for the next generation of memory technology, but its significant non-uniformity issue must be resolved for commercialization[13–17]. The variability is due to the stochastic nature of the switching mechanism. On the other hand, the stochastic phenomena in memristors are being exploited for security and computing primitives. Hardware security applications, such as true random number generators (TRNGs) and physically unclonable functions (PUFs), have been demonstrated by the inherent stochasticity of memristors[18–23]. The stochastic source can also be harnessed for computing approaches, such as stochastic neural networks[24,25]. This work suggests another utilization of the stochastic property of a diffusive memristor as the p-bits in p-computing.

A diffusive memristor is a two-terminal ionic device with a volatile threshold switching (TS) behavior. It switches to an on (TS-on) state at a specific threshold voltage and relaxes back to an off (TS-off) state upon the voltage removal. A diffusive memristor has been adopted for various applications, such as biological neurons[26], TRNGs[19–22], PUFs[23], and sensory circuits like a nociceptor[27]. It has outstanding device performance regarding power consumption, scalability, switching

---

[1]Department of Materials Science and Engineering and Inter-University Semiconductor Research Center, Seoul National University, Gwanak-ro 1, Daehag-dong, Gwanak-gu, Seoul 08826, Republic of Korea. [2]These authors contributed equally: Kyung Seok Woo, Jaehyun Kim. ✉e-mail: cheolsh@snu.ac.kr

**Table 1 | Comparison of different computation methods**

| Computation methods | Classical computing | Quantum computing | Probabilistic computing |
|---|---|---|---|
| Data expression | 0 or 1 deterministic values | Superposition of 0 and 1; an infinite number of states between 0 and 1 | Probabilistic 0 or 1 |
| Hardware implementation | CMOS-based digital logic circuits | Computing system based on electron spin resonance | Oscillating digital outputs based on stochastic devices |
| Output | Deterministic | Probabilistic | Probabilistic |
| Power consumption | High | High | Low |

speed, and endurance[16,28]. Its switching mechanism is based on a metal ion-based filamentary conducting mechanism. When a positive bias is applied to an active electrode, the metal ions generated from the active electrode migrate through an insulating layer to form a weak conductive filament (CF) between the active electrode and a passive electrode. The TS device differs from the electrochemical metallization cell, in which the metallic CF was strong enough to remain intact for a sufficiently long time, ensuring its feasible memory performance. By contrast, the metallic CF in the TS device, or diffusive memristor, is weak enough to be ruptured by even thermal energy at room temperature, driven by the high interface energy effect[29]. Therefore, careful control of (minimal) injection of the metal ions into the insulator (or electrolyte) layer is crucial to fabricating a viable TS device. Various materials have been investigated for the active electrode, such as Cu, Ag, Ni, and Ru[13,27,30,31]. However, the pure metal active electrode generally injects too many metal ions during the on-switching step, hindering reliable TS device fabrication. Recently, a Cu alloy was introduced as the active electrode, showing a promising TS behavior[22,32,33]. The $Cu_xTe_{1-x}$ active electrode shows both volatile TS behavior and non-volatile resistance switching (RS) behavior by controlling the number of Cu ions drifting out of the electrode. This is because of the lower activity of the active elements in the alloy electrode, enabling a more controlled injection of the metal ions when the voltage is applied.

This work suggests a p-computing scheme using a $Cu_{0.1}Te_{0.9}$/$HfO_2$/Pt (CTHP) diffusive memristor as the p-bit generation element. Numerous studies have reported the utilization of memristors as synaptic devices in hardware neural networks or neuromorphic circuits, but they do not require Boolean logic operations. On the other hand, p-computing pursues the Boolean logic operation, and thus, is closer to the in-memory logic operation. All 16 Boolean logic operations were implemented in this study by designing a p-computing system based on the CTHP memristor-based p-bits.

## Results

### Theoretical model of p-computing

The p-computing network is an energy-based network that updates the network to minimize the total energy. The main idea of the probabilistic network stems from quantum annealing[6]. In quantum computing, the system energy, consisting of lattice sites with spin states $x_i$, is explained by the Ising model, which was originally proposed to describe a ferromagnetic phase[34]. A summary of the Ising model for a quantum system is presented in Supplementary Note 1. The operation principle of p-computing is derived from the theoretical background of quantum annealing, which shares similar features with a stochastic Hopfield network[35]. Quantum annealing is based on an energy-based model to solve combinatorial optimization problems. In quantum annealing, the energy of a quantum system comprised of "qubits" is defined as Hamiltonians, further divided into the initial and final Hamiltonian. The initial Hamiltonian denotes the initial ground state of the system, where each qubit remains in a state with the quantum superposition of 0 and 1. As the system undergoes the annealing procedure, the initial Hamiltonian slowly develops into the final Hamiltonian, which provides a low-energy solution to the given

problem[36]. Similarly, for p-computing, each qubit can be substituted with multiple "p-bits," in which the binary states fluctuate with time[37,38]. For simplicity, the equations of a stochastic Hopfield network or a Boltzmann machine[39] are introduced, which may provide the conceptual framework of the present p-computing principle.

Figure 1a shows the architecture of the Hopfield network. The energy function of the Hopfield network follows that of the Ising model as follows.

$$E = -\sum_{j \neq i} w_{ij} x_i x_j + \sum_i \theta_i x_i \tag{1}$$

$$-\frac{\partial E}{\partial x_i} = \sum_{j \neq i} w_{ij} x_j - \theta_i, \tag{2}$$

where $x_i$ and $x_j$ are the neuron outputs with values of either 0 or 1, $w_{ij}$ is the fixed weight value connecting the two neurons, and $\theta_i$ is the applied bias. The neuron output is updated to minimize the energy function of the system for a given input, where the direction of the update is determined by the partial derivative of the energy by $x_i$, expressed in Eq. 2. If the input exceeds the threshold $\theta_i$, or Eq. 2 is greater than 0, $x_i$ increases to 1 to lower the energy. Equation 2 is also related to the input ($z_i$) to the $i$th neuron. The input into the binary stochastic neuron is processed to output $x_i$, which can be written as

$$x_i = u[\sigma(z_i) - rand], \tag{3}$$

where $u(x)$ is the step function for binary output update, $\sigma(x)$ is the sigmoid function, $z_i$ is the input into the neuron $i$, and $rand$ is the random number uniformly distributed between 0 and 1[10,11]. The $rand$ term gives stochasticity to the output, granting the probability to fire even with small $z_i$ values.

As shown in Fig. 1b, the inputs and outputs of p-bits are similar to a stochastic Hopfield network and are defined as follows:

$$E = -\sum_{j \neq i} S_{ij} p_i p_j + \sum_i \theta_i p_i \tag{4}$$

$$I_i = I_0 \left( S_{ij} p_j + \theta_i \right) = -\frac{\partial E}{\partial p_i} \tag{5}$$

$$p_i \approx u[\sigma(I_i) - rand], \tag{6}$$

where $I_i$ is the $i$th p-bit input, $p_i$ is the $i$th p-bit output, and $I_0$ is an arbitrary parameter for keeping the calculated p-bit input voltages in the valid p-bit window[9]. Here, the synaptic connection $S_{ij}$ corresponds to $J_{ij}$ of the Ising Hamiltonian. However, $S_{ij}$ is not limited to the connection weight between just two p-bits; it can be extended to the connection weight between 3 (e.g., $S_{ijk}$) or more p-bits, depending on the applications. Table 1 shows the comparison between the p-computing and other computation methods[5,9,40]. Classical computing is deterministic because the data is represented as discrete '0' or '1'.

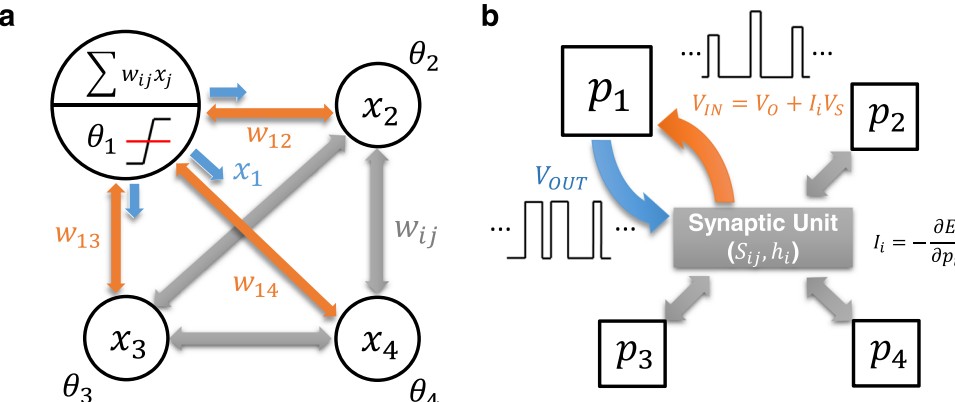

**Fig. 1 | Architecture of the computing networks. a** Network architecture of a stochastic Hopfield network. **b** Network architecture of a probabilistic network.

The quantum state is expressed by the superposition of '0' and '1' bits, making the outputs probabilistic. On the other hand, in p-computing, the data is represented as the probability of output $p_i$ fluctuating with time between '0' and '1'. Compared to quantum computing, which requires cryogenic temperature operation, p-computing is more energy-efficient because it can operate at normal temperature.

Another essential feature of p-computing is that the result is not fixed into a deterministic output. Instead, the outputs fluctuate between several energy function values; the most probable output, the output with the minimum energy function, is chosen as the result. The main difference between p-computing and machine learning is that the p-computing can calculate the results in one shot without training the weights at the expense of the loss of computing accuracy. In contrast, machine learning takes multiple training epochs, which consumes more power. However, machine learning can perform the tasks more accurately by optimizing the weight matrix.

**Memristor-based p-bit and p-computing system**

Figure 2a shows the current-voltage (I-V) curves of the TS behavior of the CTHP diffusive memristor with a compliance current ($I_{cc}$) of 10 nA. Details of the device fabrication have been reported elsewhere[22]. A $10 \times 10$ μm² electrode area of the CTHP device was fabricated in a cross-point structure, as shown in the scanning electron microscopy (SEM) image (Supplementary Fig. S1a). The structure of the CTHP device was confirmed using transmission electron microscopy (TEM) and a depth profile using Auger electron spectroscopy (AES) (Supplementary Fig. S2b, c). The TS behavior of the $Cu_xTe_{1-x}$-based memristor was observed only at x = 0.1.

Depending on the level of $I_{cc}$, which controls the number of Cu atoms forming the CF, the device can exhibit either TS or RS behaviors[41–43]. A weak CF is formed at low $I_{cc}$, so the Cu atoms composing the CF diffuse away from the weak CF quickly when the voltage is removed. A thicker CF is formed at high $I_{cc}$, turning the device into the RS mode, which is irrelevant to this study. After an electroforming process at 4.0 V, the TS behavior was confirmed in 20 consecutive sweeps with the threshold voltage ($V_{th}$) ranging from 1.5 to 2.5 V. The CTHP device achieved high cycling endurance (>$10^6$), as shown in Supplementary Fig. S1e. The endurance of Cu-based threshold switching can reach $10^{10}$ cycles, showing the potential for stable bit generation[44,45]. However, its threshold switching performance could be frustrated by changing it to the resistive switching mode, accompanied by the excessive Cu atom migration into the insulating layer.

The inset of Fig. 2b shows the circuit design of the memristor-based p-bit unit, consisting of a CTHP memristor and a comparator (HA17393, Renesas, Japan). Figure 2b shows the averaged $V_{out}$ as a function of $V_{in}$ at different voltages, and the plot follows a sigmoidal fitting curve. 500 samplings, shown in Fig. 2c, are averaged for each

data point. Figure 2c shows the comparator output voltage ($V_{out}$) as a function of time at $V_{in}$ (input pulse voltage applied to the memristor) = 5.10 V, 5.23 V, and 5.32 V. The $V_{in}$ duration was fixed to 150 μs with 25 μs leading and trailing times. A rest time of 800 μs was given to allow the memristor to relax fully. At higher $V_{in}$, the device is more likely to switch to a low resistance state. This property results in a high $V_{out}$ occurring more frequently. In this case, the critical property of the p-bit is that the $V_{in}$ can control the probability of device switching (or p-bit state) following the sigmoidal function. This behavior is similar to the stochastic neuron in Eq. 6; the $n$th p-bit output for the circuit parameters ($p_n = \frac{V_{OUT}}{V_{DD}}$) can be written in the following form:

$$\frac{V_{OUT}}{V_{DD}} \approx u\left[\sigma\left(\frac{V_{IN} - V_O}{V_S}\right) - rand\right],\qquad(7)$$

where $V_{DD}$ is the supply voltage of the comparator, $V_O$ is the offset voltage at which the probability of output 'high' is 50%, which was ~5.23 V in this work, $V_{IN}$ is the input voltage into the circuit, and $V_S$ is the scaling voltage[35]. $\frac{V_{IN} - V_O}{V_S}$ is the normalized input, $I_i$, which determines the probability of the p-bit state (Supplementary Fig. S2).

Still, the bit generation speed of memristor-based p-bits can be further improved through device engineering. Substituting the insulator with a higher Cu ion diffusivity can further accelerate the CF formation and dissolution processes[29]. The switching time of memristors can be as short as tens of picoseconds, showing the potential for fast and low-power computing[18,46]. The electrode structure of a memristor is much simpler than the MTJ. They also have a larger tolerance for the thickness variation of the insulating layer compared with the extremely tight allowable thickness of the insulating layer in MTJ.

**Logic operations**

Logic operations can be executed with the p-computing network based on the memristor-based p-bits. For instance, an 'AND' operation returns 'true' when all the inputs are 'true.' Otherwise, the output is false. The corresponding equation that satisfies these conditions is written as $y_1 = x_1x_2$, which is then used to create a cost function for the AND operation. The input functions are obtained by differentiating the corresponding cost functions following Eq. 5. Each variable is assigned to the p-bit; thus, a three-p-bit network is required to operate the AND logic, as shown in Fig. 3a. Similarly, all 16 Boolean logic operations can be performed with appropriate cost functions. Definitions of cost (or energy) functions and the resulting input functions for all 16 Boolean logic operations are shown in Supplementary Table 1. The cost function of AND logic, for example, is given as the square of the difference between the true value ($x_1x_2$) and current value ($y_1$), which is

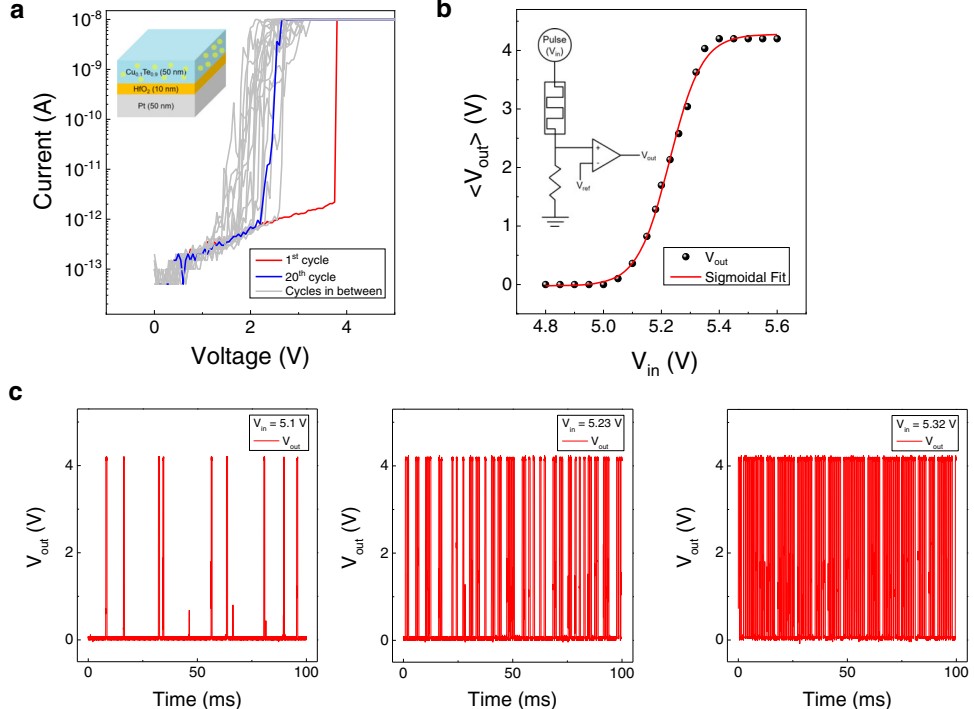

**Fig. 2 | A p-bit demonstration using a CTHP memristor. a** I-V curves of the CTHP memristor with a schematic of the CTHP memristor shown in the inset. **b** Averaged $V_{out}$ as a function of $V_{in}$ with a sigmoidal fitting curve. The inset shows a circuit diagram of the CTHP-based p-bit. **c** $V_{out}$ of the CTHP-based p-bit at $V_{in}$ = 5.1 V, 5.23 V, and 5.32 V.

similar to how the cost function is defined in deep learning of neural networks. When the cost function of the AND logic is fully expressed, $E(x_1, x_2, y_1) = x_1 x_2 - 2x_1 x_2 y_1 + y_1$, there are multiplications and summations of the inputs, $x_1, x_2$, and output, $y_1$, with the relevant coefficients, 1, −2, and 1. The coefficients define the connection strength between the p-bit outputs, which is analogous to the role of synaptic weights connecting the neurons in machine learning. When different logic gates are necessary, these functional relationships between the terms and relevant coefficients should be modified. Supplementary Table 1 summarizes all these relationships and coefficients for the 16 Boolean logic gates. As shown later, even complex gates, such as full adder, can be defined similarly. Besides, there is a crucial difference between synaptic weights in p-computing and neural networks. In p-computing, the synaptic weights are fixed for a given logic operation, but they evolve with training for a given task in the neural networks.

Next, more detailed explanations are given on how the p-computing can be executed. First, the input functions should be realized by networking the synaptic hardware and multiple p-bit circuits. Programmable digital circuits such as a field programable gate array (FPGA) are the most suitable approach to demonstrate such input functions with the multiplications of p-bit outputs by hardware. Figure 3b shows the schematic diagram of such hardware construction. Three p-bit circuits for $x_1, x_2$, and $y_1$, each composed of a CTHP memristor and a comparator, are connected to the inputs of FPGA, and the FPGA outputs three bits corresponding to $x_1, x_2$, and, $y_1$. The FPGA is programmed to output the correct bits depending on the given logic operations using the input and cost functions. In this work, all logic operations were implemented by simulation based on the CTHP-based p-bit characteristics and the cost functions. The simulation was performed using the fitted sigmoid relation and parameters calculated from Eq. 7. The sigmoid fitting curve in Fig. 2b is based on the averaged $V_{out}$, but variations exist, as shown in Fig. 2c. The widest distribution is found at 5.23 V, the point at which the p-bit exhibits the most stochastic behavior. As the $V_{in}$ value deviates farther from 5.23 V, the

distributions become narrower, and the p-bit becomes deterministic to '0' or '1.' Since the memristor always has variability issues, such as cycle-to-cycle and device-to-device variations, these variations were considered for all logic operations in the simulation. For each clock cycle, a random output of 0 or $V_{DD}$ is generated from the comparator. This output is normalized to 0 or 1 by the relation, $p_n = \frac{V_{OUT}}{V_{DD}}$.

For the forward operations, the input voltages into the p-bits, corresponding to $x_1$, and $x_2$, are derived from Eq. 7 by $V_{IN} = I_i V_S + V_O$. When the inputs are 0, $V_{IN,x1}$ and $V_{IN,x2}$ are fixed to sufficiently low voltage, ca. 5.10 V, to ensure switching probability close to 0. Under this circumstance, the p-bit circuits for the two inputs most frequently output zero voltage, which drives the FPGA to output the corresponding bit of 0. For input 1, the $V_{IN}$ value of the corresponding input p-bit circuit is settled to 5.32 V, which renders the FPGA mostly produce the corresponding bit of 1. Next, the corresponding $y_1$ value must be determined for the given inputs. For this operation, the $V_{IN,y1}$ is initially settled to $V_O$ (-5.23 V in this case), which is the voltage of 50% switching probability, then it is floated. Next, the $V_{IN,y1}$ must be changed to a value, which can represent the AND logic operation. By the definition of the input function of AND logic, $I_{y1} = 2x_1 x_2 - 1$, $I_{y1}$ is calculated to be −1 for $x_1$ or $x_2$ = 0. In this case, the switching probability is ∼27% (See Supplementary Fig. S2). $V_{IN,y1}$ is then calculated to show the $y_1$ p-bit outputs $<\frac{V_{OUT}}{V_{DD}}>$ ∼27%, and is inputted to the $y_1$ p-bit. Under this circumstance, the FPGA outputs the $y_1$ bit mostly 0. However, it should be noted that there is a significant chance for the output $y_1$ bit of FPGA is 1 due to the involvement of CTHP variation and *rand*. Therefore, when the procedures discussed above are simulated 100 times, the probability of the outputs of the FPGA $(x_1, x_2, y_1)$ to be (000) and (001) are ∼0.84 and ∼0.16, respectively, as shown in the left panel of Fig. 3d (i, forward operation). The corresponding probabilities for (010), (011); (100), (101); (110), (111) are ∼0.84, ∼0.16; ∼0.84, ∼0.16; ∼0.25, ∼0.75, indicating that the correct AND logic operations are acquired. However, it can be argued that one of the fundamental assets of any logic operation, i.e., logic correctness, is only probabilistically confirmed. Therefore, it can be questioned the merit of such logic gating

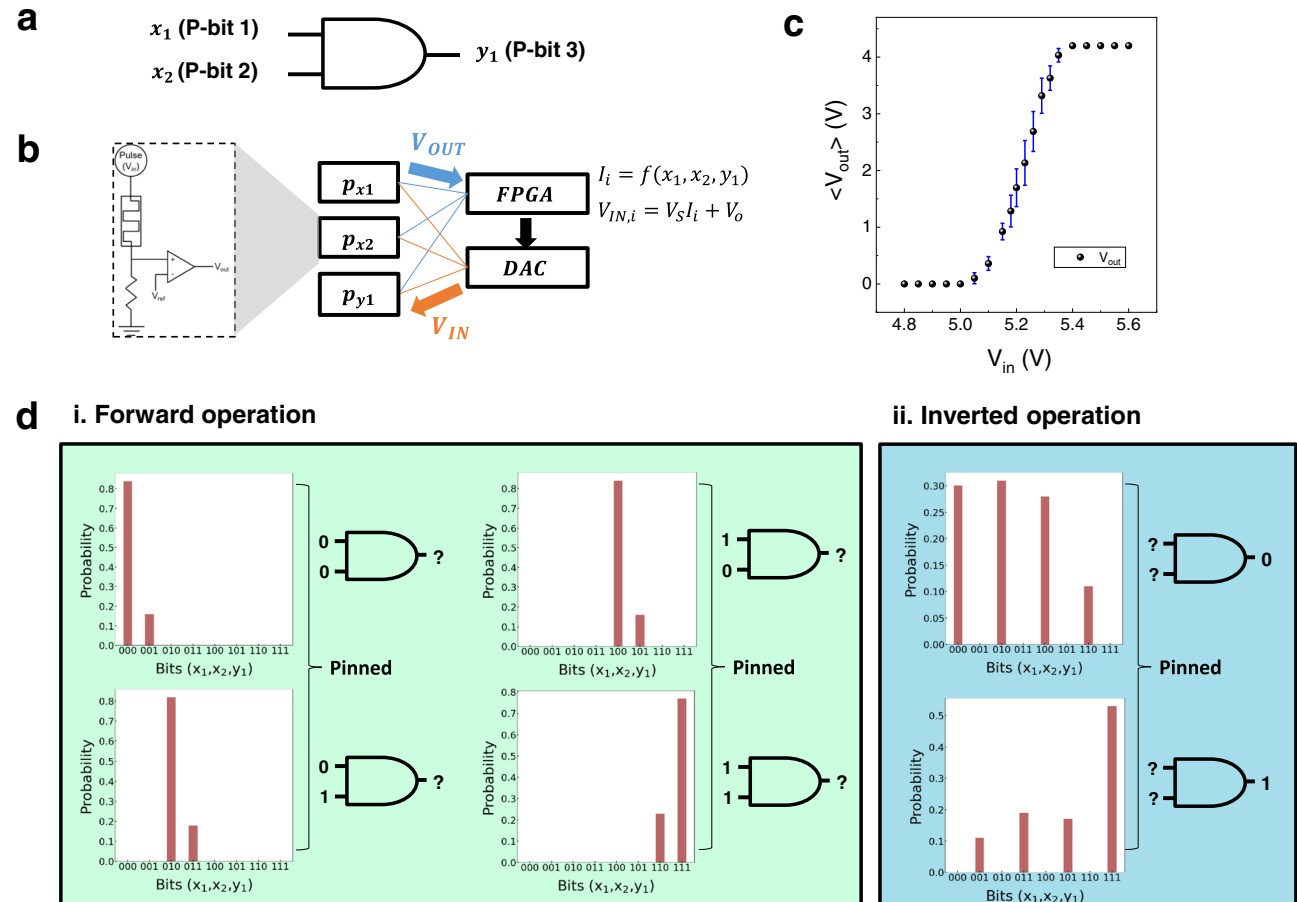

**Fig. 3 | Boolean logic operation through the memristor-based p-computing. a** A schematic AND gate using three p-bits. **b** Hardware implementation using an FPGA and three p-bits. **c** Distributions of the memristor-based p-bit at different input voltages (4.8 to 5.6 V). **d** Three-p-bit AND operation with the forward (left) and inverted AND operations (right).

using the p-bits. One of the reasonable rationales is the invertible calculation. The following inverted calculation can be performed using the same hardware for AND logic. In this case, $y_1$ is given first to be 0 or 1, which then requires $x_1, x_2 = (0,0), (0,1), (1,0)$ or $(1,1)$. For this operation, the $V_{IN,y1}$ is fixed to low (5.10 V) or high (5.32 V) voltage, while the $V_{IN,x1}$ and $V_{IN,x2}$ inputs are first settled to 5.23 V and then floated. Then, $V_{IN,x1}$ and $V_{IN,x2}$ values are determined based on their respective input functions, $I_{x1}$ and $I_{x2}$. Subsequently, similar procedures are repeated to determine the $x_1, x_2$ for the given $y_1$ value. The right panel of Fig. 3d (ii, inverted operation) reveals that (000), (010), and (100), corresponding to the correct case for $y_1 = 0$, have a probability of ~0.3, whereas other incorrect cases have a probability of < ~0.1. For the $y_1 = 1$, the correct and incorrect cases have their respective probability of ~0.5 and < ~0.2. Therefore, it can be inferred that the inverted AND logic operation could be feasibly (statistically) achieved using the given p-bit circuits. The supplementary information (Supplementary Figs. S3–S17) also shows that inverted logic operations for all the remaining Boolean gates are possible.

In the field of memristor-based logic-in-memory (LIM), several studies have implemented such logic operations[47]. However, the non-uniformity issues in the memristor's performance adversely affect the accuracy of the logic operations. In the case of memristor-based p-bit computing, in contrast, the calculation results are less prone to error by the diverse variability since the answer is found probabilistically. Also, any calculation is possible in a one-shot method with a suitable cost function for a particular operation. The following section discusses this crucial feature.

## Complex operations

Memristor-based p-computing can function as an arithmetic logic unit (ALU), a digital circuit for arithmetic operations. Although the complementary metal-oxide-semiconductor (CMOS)-based ALU can perform complex operations, drawbacks arise in power consumption, circuit complexity, scalability, and operation speed. On the other hand, memristor-based p-bit computing can perform any functions without the drawbacks above. For instance, a 1-bit half-adder operation was performed in a four-p-bit network, as shown in Fig. 4a. With the cost function for the half-adder, both forward and reverse operations were achieved using four p-bits (each for two inputs, sum, and carry) in Fig. 4b. The half-adder can be extended to the full adder, which can be implemented in a five-p-bit network by simply adding a p-bit representing a bit carried in from the previous operation (Supplementary Fig. S18). A more complex 2-bit by 2-bit binary multiplier operation is shown in Fig. 4c, which can demonstrate the feasible operation. The reverse operation of this multiplier is factorization, which can efficiently be executed in the same manner. More p-bits are required for larger integers in binary form.

Unlike CMOS-based ALU, logic cascading is unnecessary in the p-computing for many complex logic operations. They are possible in a one-shot method as long as the cost function is given. However, the probabilistic approach has a problem because the criterion for the correct answer may be unclear. If there is one solution for a specific operation, the highest probability can be set as the answer. On the other hand, when there are two or more solutions, it can be problematic to set the standard for the correct answers.

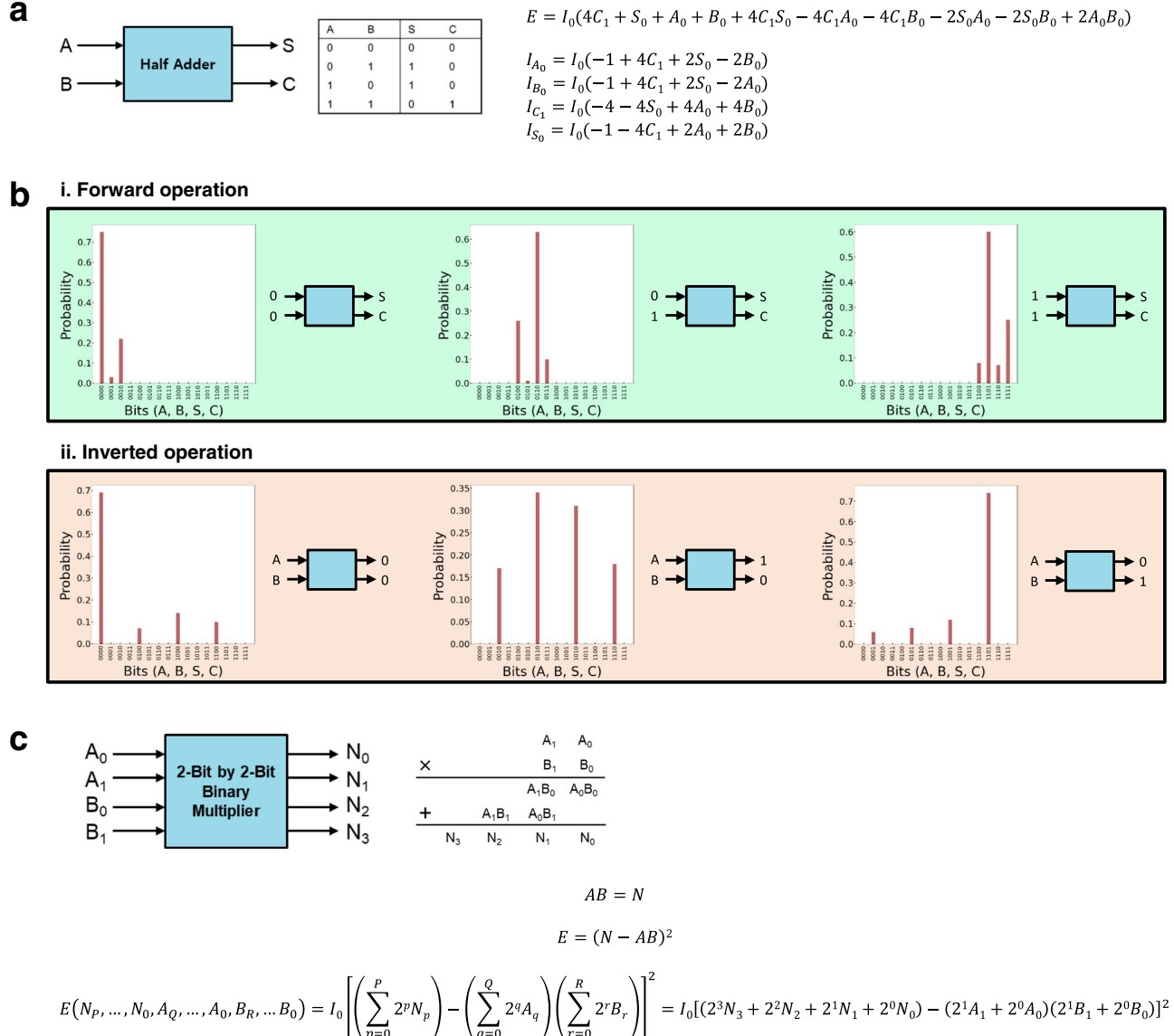

**Fig. 4 | Complex operations through the memristor-based p-computing. a** A half-adder using a four-p-bit network and its input functions. **b** Forward and inverted half-adder operations with $I_0 = 1$. **c** Design for a 2-bit by 2-bit binary multiplier using an eight-p-bit network.

Another potential problem could arise. For highly complex logic operations, the number of involved memristors should be increased accordingly. In this case, a too high cell-to-cell variation could induce a malfunction. Furthermore, too large fan-in and fan-out could be another problem, which may increase the overhead of driving circuits. Therefore, these aspects must be considered carefully.

Since p-computing is conceptually related to machine learning and quantum annealing, its applications can be expanded into these fields. As the behavior of the p-bit is similar to that of the stochastic neuron, it can be utilized to implement Bayesian inference and the Boltzmann machine in a probabilistic framework[48–50]. Adopting the theoretical background of quantum annealing, p-computing can efficiently solve optimization problems, such as traveling salesman problems[51].

## Discussion

This study proposes a p-computing scheme using the CTHP diffusive memristor. The theoretical model of p-computing resembles the Boltzmann machine, which is based on the Ising model of a recurrent neural network. The p-computing can execute the Boolean logic operations and produce results in one shot without training the weights. The data in p-computing is stored as the p-bit, which has the probability of being '0' and '1'. The stochastic behavior of the CTHP memristor successfully demonstrated the p-bit property.

Moreover, the cost and input functions for all 16 Boolean logic operations were derived in a more straightforward form than the previous works. All logic operations were implemented in forward and reversed directions through the memristor-based p-computing network. Complex functions, such as full adder and multiplication/factorization, were also suggested, showing the methodology's potential to be applied to more complex logic circuits. Finally, a comparison between other p-computing hardware and this work is shown in Table 2[18,46,52,53]. The average power consumption of the p-bit circuit was calculated using the pulse output of the device (Supplementary Fig. S19). Based on a 2-terminal metal-insulator-metal structure, p-bits built with memristors are advantageous in area efficiency and production cost compared with other p-bits when integrated into a larger network. The memristor-based p-bit demonstration shows promise for

**Table 2 | Comparison between various probabilistic computing hardwares**

| | CMOS-based[12] | Magnetic Tunnel Junction-based[12, 53] | Memristor-based |
|---|---|---|---|
| Fabrication (Device) | Complex (Transistor Logic) | Complex (Ta/Pt/[Co/Pt]$_7$/Co/Ru/[Co/Pt]$_2$/Co/Ta/CoFeB/ MgO/CoFeB/Ta/Ru/Ta) | Simple Metal/Insulator/Metal (Cu$_x$Te$_{1-x}$/ HfO$_2$/Pt) |
| Power Consumption | 200 μW | 10 μW | 154 nW @ 5.4 V |
| CMOS circuits (Number of transistors) | LFSR (1194) | NMOS+Comparator (11) | Comparator (10) |
| Potential switching speed of the device | 100 ps[12] | 200 ps[18, 52] | 85 ps[18, 46] |

For the MTJ and CTHP devices, only energy for the probabilistic device was considered (comparator energy consumption was excluded). However, the operation speed of the MTJ is higher than CTHP.

computing hardware using diffusive memristors, which may overcome the memory wall issue of the current von Neumann computing method.

## Methods

### Memristor fabrication

To fabricate a cross-point structure of the CTHP device, an 8 nm thick Ti adhesion layer and a 50 nm thick Pt bottom electrode (BE) were deposited on a SiO$_2$/Si substrate using an electron beam evaporator (SRN-200, SORONA), followed by a lift-off process. Then, a 10 nm thick HfO$_2$ insulating layer was deposited using thermal atomic layer deposition (Plus 200, CN-1 Co.) using Hf[N(CH$_3$)(C$_2$H$_5$)]$_4$ and O$_3$ as Hf precursor and oxygen source, respectively at a 280 °C substrate temperature. A 50-nm-thick Cu$_{0.1}$Te$_{0.9}$ top electrode (TE) was DC-sputtered by co-sputtering from Cu and Te targets (07SN014, SNTEK) with the power of 10 W and 120 W, respectively. Finally, a 40 nm thick Pt passivation layer was deposited using the electron beam evaporator, followed by the lift-off process. The DC electrical characterizations were conducted using a semiconductor parameter analyzer (HP4145B, Hewlett-Packard). An Agilent 81110 A pulse generator was used for the pulse measurements. During the electrical measurements, the TE was biased, and the BE was grounded. The p-bit circuit was built on a breadboard consisting of a memristor and a comparator. The cross-point structure was confirmed using an SEM (S-4800, Hitachi) image. The depth profile and the cross-sectional image of the device were acquired using AES (PHI-700, ULVAC-PHI) and TEM (JEM-ARM200F, JEOL), respectively.

## Data availability

All the relevant data are available from the corresponding authors upon reasonable request.

## Code availability

Computational results were obtained using Python software. Python was also used to perform the logic operations. All the relevant codes are available from the corresponding authors upon reasonable request.

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

## Acknowledgements

This work was supported by the National Research Foundation of Korea (NRF) grant [2020R1A3B2079882].

## Author contributions

K.S.W. designed the study concept and approach and performed electrical measurements. J.K. fabricated the devices and refined the concept. J.H. contributed to the Python simulation. W.K. contributed to the device analysis. Y.H.J contributed to the power calculation. K.S.W. and J.K. analyzed the results and wrote the manuscript draft. C.S.H. supervised the entire project and wrote the final manuscript.

## Competing interests

The authors declare no competing interests.
