## [Peer Review File · Nature Communications]

REVIEWER COMMENTS

Reviewer #1 (Remarks to the Author):

In this manuscript the authors demonstrate a hardware implementation of probabilistic circuits using the stochastic switching behavior of a diffusive memristor. The implementation is interesting and the paper has the potential to be an important alternative compared to other realizations of stochastic nanodevices for p-computing, however, the manuscript in its current form needs significant work to be publishable in any venue.

First, the comparisons to quantum computing that are in the text are quite inaccurate and careless. For example, the statement: “The principle of p-computing is derived from the theoretical background of quantum computing and Hopfield network-based machine learning” is not specific enough to make sense. It is true that p-computing has a lot of similarities to a type of quantum computing (namely quantum annealing pursued by D-Wave) but without this qualification it is inaccurate. Second, the authors repeated use of “Hopfield Networks” is also inaccurate because hardware p-circuits most naturally resemble Boltzmann Networks (that are admittedly stochastic Hopfield Networks but without the unnecessary connotations to associative memory etc., which is the main focus of Hopfield Networks).

Similarly, Table I provides little comparison to anything substantial other than making perfunctory statements about differences between p and q-computing. What would have been a far more useful comparison in Table I would be to compare the authors' implementation in this paper with respect to Magnetic Tunnel Junction based p-bits which have been explored and demonstrated extensively.

For example, could the fluctuations observed in diffusive memristors can reach GHz time scales such as those achieved by the IBM [Safranski et al, Nano Letters 2020] and Tohoku Groups [Hayakawa et al, PRL 2020] in magnetic tunnel junctions? How should the power dissipation of their p-bit compare to the MTJ-based p-bit evaluated as dissipating 20 microWatts per p-bit discussed in Hassan et al, Phys. Rev. Applied 2021? Unfortunately, crucial comparisons like these are replaced by somewhat artificial comparisons to quantum computing without much support.

Similarly, unless I missed it, there is not enough detail regarding how the synaptic operation is performed, even in the lengthy supplementary.

In short, I believe the manuscript has potential, but is in need of significant and careful revision, especially in regards to how quantum computing references. Careful comparisons with actual state of the art is needed. More details regarding the scheme needs to be presented.

Reviewer #2 (Remarks to the Author):

Probabilistic computing, or computing using "p-bits" originally proposed by Datta group from Purdue University has gained some traction as one of the approaches to implement an Ising Machine, which themselves are investigated for solving certain problems that can be cast or recast as a global optimization problem. A p-bit computer is "wired" up corresponding to the optimization program and then the result is obtained by either through simulated annealing or via very long term sampling of the p-bit values which seeks to obtain the bit pattern corresponding to the ground-state (lowest eigen value) of the Hamiltonian being encoded by the p-bit network. As such this form of computing is suitable for any probabilistic algorithm that is based on energy minimization and a nano-material that shows significant stochastic electrical response is well suited to provide the stochastic samples necessary for the operation of such an optimizer.

Datta group has presented quite a few papers on the theoretical analysis, as well as low barrier magnet MTJ based hardware (published in Nature), as well as emulated p-bits generated purely algorithmically using CMOS circuits, including microcontrollers, and FPGA (from Camsari group at UCSB). It should also be noted that p-bits do not provide the only implementation for an Ising machine solver as many other groups have approached this problem using dynamical systems built from oscillators built from CMOS, variety of memristors, optical substrates, charge density waves in 2D materials etc. It should also be noted that a significant number of quantum computing research, most prominently D-Wave, are solving the same problem via quantum methods (quantum annealing ratehr than simulated annealing) even though the problem itself is classical and small problems can be solved using simple MATLAB codes on a mid-range laptop. Therefore the jury is still out if this method has any intrinsic benefits over conventional computing and at what problem sizes might such an advantage show up.

The authors have presented a work that attempts to build proof-of-concept Ising chains using a Copper-Telluride-Platinum Hafnia (CTHP) diffusive memristor. They have fabricated p-bits using a combination of the CTHP, a resistor, and a comparator. This is in lines of other designs presented using MTJs. The circuits were assembled on a breadboard, which as also as per the standard

practice. Therefore I find the methodology section as per the standard setups being used in this area.

However my central concerns are against the presentation of the material, going beyond the stylistic approaches. The authors present the p-bit with a comparison between classical/deterministic, p-computing and quantum computing. This comparison starts right at the very beginning, e.g. line 34-39. While the quantum computing has indeed being used in Ising machine like problem as I mentioned before, this contentions is not warranted. Quantum computing's power arises from large scale entanglement which really forms a large state space to compute over. Similarly quantum computers often exploit "negative" probability amplitudes that allow for destructive interference of the compute possibilities (see Grover search or the classic Deutsch algorithm that started off the field). I suspect that authors are motivating the p-bits as a "poor man's quantum computer" since quantum computers are significantly harder to build and operate. However a much simpler motivation of solving BPP algorithms, going beyond P ones would have been sufficient. There is also a strong conjecture that $P=BPP$ which, if true, will render such accelerators moot. Therefore the table 1 is not illuminating beyond trivial comparison.

Another such concerning point is the comparison made between machine learning methods and p-computing. It is puzzling to me because authors themselves correctly point out that p-computers are merely stochastic Hopfield networks, a very popular neural network, also called the Boltzmann machine. The operating principles of operation is precisely the same, energy minimization and a Boltzmann distribution based programming of energy eigen values. Even the original proponent of p-bit have clearly stated that their approach to actually figure out the weights depend on the same algorithm that is well known and used in Hopfile d network literature, i.e. carefully aligning the eigenvectors and thus the p-bit values to the truth table being programmed. For an unknown problem, some sort of weight update algorithm will be necessary even for p-computing. All this makes me wonder how much thought the authors have given in connecting these established and well known pieces in their own minds, and this reflects in the exposition.

My last concern is regarding the novelty of the work. The basic principles are well known and established in the lietrature, only difference is that they have used a particular material to create their p-bits as a drop-in replacement for a BSN which again is a very old and well known concept, see the very first chapter of Simon Haykin's celebrated book on NNs. I am not sure that such an incremental work warrants a publication at a venue such as Nature Communication, given the utter lack of surprise in the presented results which do not illuminate or advance the field in any significant way.

Reviewer #3 (Remarks to the Author):

I think this manuscript reports significant new results using memristive elements in a role that is very different from the usual synaptic functions. As such I believe it should be published in a journal like Nature Communications after appropriate revisions.

There are statements in this paper that do not sound right to me. For example, in the context of quantum computing I see the statement: ".. lacking the general algorithm for Boolean logic operations is another critical huddle for its widespread use." I believe that at least some of the probabilistic algorithms demonstrated here (like Boolean operations and factorization) were first explored in the context of adiabatic quantum computing.

One minor comment is about their use of "Hopfield networks" to describe what is probably a "Boltzmann machine" which is a stochastic Hopfield network.

Since this paper uses memristors to implement a function that has been implemented with magnetic tunnel junctions, it seems to me that a paragraph comparing them may be of interest to readers.

We appreciate the valuable and constructive comments from the reviewers, which contributed greatly to enhancing the quality of the manuscript. (NCOMMS-22-02193) We did our best to comply with each comment and correct the manuscript accordingly. Our point-by-point responses to each comment are shown below. In addition, the related modifications are highlighted and also applied to the revised manuscript.

REVIEWER COMMENTS

Reviewer #1 (Remarks to the Author):

In this manuscript the authors demonstrate a hardware implementation of probabilistic circuits using the stochastic switching behavior of a diffusive memristor. The implementation is interesting and the paper has the potential to be an important alternative compared to other realizations of stochastic nanodevices for p-computing, however, the manuscript in its current form needs significant work to be publishable in any venue.

→ Answer from authors:

We appreciate the brief overview and concise summarization of the manuscript by the reviewer. Below are the point-by-point responses to each comment.

Comment 1: First, the comparisons to quantum computing that are in the text are quite inaccurate and careless. For example, the statement: “The principle of p-computing is derived from the theoretical background of quantum computing and Hopfield network-based machine learning” is not specific enough to make sense. It is true that p-computing has a lot of similarities to a type of quantum computing (namely quantum annealing pursued by D-Wave) but without this qualification it is inaccurate.

→ Answer from authors:

We thank the reviewer for these valuable comments. We revised the introductory part to make it more specific to the relevant context of quantum computing (quantum annealing). We originally thought this was a bit too specific, but now we agree with the reviewer’s comments and modified the text.

→ Page 5, deleted:

~~The working principle of the p-computing network can be more directly derived from the Hopfield~~

network. The Hopfield network is a single-layer neural network with neurons corresponding to the spins on the lattice sites of the Ising model, fully connected to each other.³⁵ The weight represents the strength of each connection.

→ Page 5, added:

The operation principle of p-computing is derived from the theoretical background of quantum computing called quantum annealing, which shares similar features with a stochastic Hopfield network.³⁵ Quantum annealing is based on an energy-based model to solve combinatorial optimization problems. In quantum annealing, the energy of a quantum system comprised of “qubits” is defined as Hamiltonians, further divided into the initial and final Hamiltonian. The initial Hamiltonian denotes the initial ground state of the system, where each qubit remains in a state with the quantum superposition of 0 and 1. As the system undergoes the annealing procedure, the initial Hamiltonian slowly develops into the final Hamiltonian, which provides a low-energy solution to the given problem.³⁶ Similarly, for p-computing, each qubit can be substituted with multiple “p-bits,” in which the binary states fluctuate with time.^{37,38} For simplicity, the equations of a stochastic Hopfield network or a Boltzmann machine³⁹ are introduced, which may provide the conceptual framework of the present p-computing principle.

Comment 2: Second, the authors repeated use of “Hopfield Networks” is also inaccurate because **hardware p-circuits most naturally resemble Boltzmann Networks** (that are admittedly stochastic Hopfield Networks but without the unnecessary connotations to associative memory etc., which is the main focus of Hopfield Networks).

→ Answer from authors:

We appreciate this careful comment. We agree that the repeated use of “Hopfield Network” may confuse the reviewer and readers. We made corrections and added a few citations regarding the hardware implementations of Boltzmann networks to prevent confusion.

→ Page 5, modified:

The operation principle of p-computing is derived from the theoretical background of quantum computing called quantum annealing, which also shares similar features with a stochastic Hopfield network.³⁵

→ Page 5, added:

For simplicity, the equations of a stochastic Hopfield network or a Boltzmann machine³⁹ are introduced, which may provide the conceptual framework of the present p-computing principle.

Comment 3: Similarly, Table I provides little comparison to anything substantial other than making perfunctory statements about differences between p and q-computing. **What would have been a far more useful comparison in Table I would be to compare the authors' implementation in this paper with respect to Magnetic Tunnel Junction based p-bits** which have been explored and demonstrated extensively.

For example, could the fluctuations observed in diffusive memristors can reach GHz time scales such as those achieved by the IBM [Safranski et al, Nano Letters 2020] and Tohoku Groups [Hayakawa et al, PRL 2020] in magnetic tunnel junctions? How should the power dissipation of their p-bit compare to the MTJ-based p-bit evaluated as dissipating 20 microWatts per p-bit discussed in Hassan et al, Phys. Rev. Applied 2021? Unfortunately, crucial comparisons like these are replaced by somewhat artificial comparisons to quantum computing without much support.

→ Answer from authors:

We thank the reviewer for this fruitful advice. Table I was added to assist the readers who may not be well familiar with quantum and probabilistic computing principles. We agree that a more realistic comparison between our work and previous works would be more appropriate to prove the novelty of the manuscript. Therefore, we organized Table II with quantitative data such as power, fabrication and size, shown below. The power consumption was calculated by the equation: $P = V_{in}I_{avg}$, where V_{in} is the input voltage, and I_{avg} is the average current flowing through the CTHP device. A typical comparator design was considered for the comparison of the cell area.

→ Page 13, added:

Complex functions, such as full adder and multiplication/factorization, were also suggested, showing the potential of the proposed method to apply to more complex logic circuits. Finally, a comparison between other probabilistic computing hardware and this work is shown in Table II.

Table II: Comparison between various probabilistic computing hardware. For the MTJ and CTHP devices, only energy for the probabilistic device was considered (comparator energy consumption was excluded). However, the operation speed of the MTJ is higher than CTHP.

	CMOS-based¹²	Magnetic Tunnel Junction-based^{12,49}	This work
--	--------------------------------	---	------------------

Fabrication (Device)	Complex (Transistor Logic)	Complex (Ta/Pt/[Co/Pt] ₇ /Co/Ru/[Co/Pt] ₂ /Co/Ta/CoFeB/MgO/CoFeB/Ta/Ru/Ta)	Simple Metal/Insulator/Metal (Cu _x Te _{1-x} /HfO ₂ /Pt)
Power Consumption	200 μW	10 μW	2.3 μW
CMOS circuits (Number of transistors)	LFSR (1194)	NMOS+Comparator (11)	Comparator (10)

Comment 4: Similarly, unless I missed it, **there is not enough detail regarding how the synaptic operation is performed**, even in the lengthy supplementary.

In short, I believe the manuscript has potential, but is in need of significant and careful revision, especially in regards to how quantum computing references. Careful comparisons with actual state of the art is needed. More details regarding the scheme needs to be presented.

→ Answer from authors:

Thank you for pointing out a missing explanation. We agree that the details about synaptic operations in the p-bit network were not specified in the main text. We have expanded the details of synaptic operations, and also rewrote the “Logic operations” section almost entirely. The logic operations can be executed by calculating the input functions in Supplementary Table 1. The input functions are composed of the coefficients and the p-bit outputs, analogous to the synaptic weights and the neuron outputs in neural networks. Therefore, it is necessary to build hardware that can perform the sums and products in the equations. For this purpose, FPGA might be the most suitable solution. However, we found that the actual setting of the FPGA required quite extensive engineering efforts, which bears little relevance to the core idea of this work. Therefore, we used a computer simulation method to emulate the computing environment, and we conducted simulations with experimental p-bit circuit results. The operation details were added below, taking AND operation as an example.

→ Page 9-10, modified:

Logic operations. With the p-computing network based on the memristor-based p-bits, logic operations can be executed. For instance, an ‘AND’ operation returns ‘true’ when all the inputs are ‘true.’ Otherwise, the output is false. The corresponding equation that satisfies these conditions is written as $y_1 = x_1x_2$, which is then used to create a cost function for the AND operation. The input functions are obtained from differentiating the corresponding cost functions following Equation 5. Each variable is assigned to the p-bit, and thus, a three-p-bit network is required to operate the AND logic, as shown in Figure 3a. Similarly, all 16 Boolean logic operations can be performed with appropriate cost functions. Definitions of cost (or energy) functions and the resulting input functions for all 16 Boolean logic operations are shown in Supplementary Table 1. The cost function of AND logic, for example, is given as the square of the difference between the true value (x_1x_2) and current value (y_1), which is similar to how the cost function is defined in deep learning of neural networks. When the cost function of the AND logic is fully expressed, $E(x_1, x_2, y_1) = x_1x_2 - 2x_1x_2y_1 + y_1$, there are multiplications and summations of the inputs, x_1 , x_2 , and output, y_1 , with the relevant coefficients, 1, -2, and 1. The coefficients define the connection strength between the p-bit outputs, which is analogous to synaptic weights connecting the neurons in machine learning. When different logic gates are necessary, these functional relationships between the terms and relevant coefficients should be modified. Supplementary Table I summarizes all these relationships and coefficients for the 16 Boolean logic gates. As shown later, even complex gates, such as full adder, can be defined similarly. Besides, there is a crucial difference between synaptic weights in p-computing and neural networks. In the p-computing, the synaptic weights are fixed for a given logic operation, but they evolve with training for a given task in the neural networks.

Next, more detailed explanations are given on how the p-computing can be executed. First, the input functions should be realized by networking the synaptic hardware and multiple p-bit circuits. Programmable digital circuits such as a field programmable gate array (FPGA) are the most suitable approach to demonstrate such input functions with the multiplications of p-bit outputs by hardware. Figure 3b shows the schematic diagram of such hardware construction. Three p-bit circuits for x_1 , x_2 , and y_1 , each composed of a CTHP memristor and a comparator, are connected to the inputs of FPGA, and the FPGA outputs three bits corresponding to x_1 , x_2 , and y_1 . The FPGA is programmed to output the correct bits depending on the given logic operations using the input and cost functions. In this work, all logic operations were implemented by simulation based on the CTHP-based p-bit characteristics and the cost functions. The simulation was performed using the fitted sigmoid relation and parameters calculated from Equation 7. The sigmoid fitting curve in Figure 2b is based on the averaged V_{out} , but variations exist, as shown in Figure 3c. The widest distribution is found at 5.23 V, where the p-bit exhibits the most stochastic behavior. As the V_{in} value deviates farther from 5.23 V, the distributions become narrower, and the p-bit becomes deterministic to ‘0’ or ‘1.’ Since the memristor always has

variability issues, such as cycle-to-cycle and device-to-device variations, these variations were considered for all logic operations in the simulation. For each clock cycle, a random output of 0 or V_{DD} is generated from the comparator. This output is normalized to 0 or 1 by the relation, $p_n = \frac{V_{OUT}}{V_{DD}}$.

For the forward operations, the input voltages into the p-bits corresponding to x_1 , and x_2 are derived from Equation 7 by $V_{IN} = I_i V_S + V_O$. When the inputs are 0, $V_{IN,x1}$ and $V_{IN,x2}$ are fixed to sufficiently low voltage, ca. 5.10 V, to ensure switching probability close to 0. Under this circumstance, the p-bit circuits for the two inputs most frequently output zero voltage, which drives the FPGA to output the corresponding bit of 0. For the input 1, the V_{IN} value of the corresponding input p-bit circuit is settled to 5.32 V, which renders the FPGA mostly produce the corresponding bit of 1. Next, the corresponding y_1 value must be determined for the given inputs. For this operation, the $V_{IN,y1}$ is initially settled to V_O (~5.23 V in this case), which is the voltage of 50% switching probability, and then it is floated. Next, the $V_{IN,y1}$ must be changed to a value, which can represent the AND logic operation. By the definition of the input function of AND logic, $I_{y1} = 2x_1x_2 - 1$, I_{y1} is calculated to be -1 for x_1 or $x_2 = 0$. In this case, the switching probability is ~27 % (See Supplementary Figure S2). $V_{IN,y1}$ is then calculated to show the y_1 p-bit outputs $\langle \frac{V_{OUT}}{V_{DD}} \rangle \sim 27\%$, and is inputted to the y_1 p-bit. Under this circumstance, the FPGA outputs the y_1 bit mostly 0. However, it should be noted that there is a significant chance for the output y_1 bit of FPGA is 1 due to the involvement of CTHP variation and *rand*. Therefore, when the procedures discussed above are simulated 100 times, the probability of the outputs of the FPGA (x_1, x_2, y_1) to be (000) and (001) are ~0.84 and ~0.16, respectively, as shown in the left panel of Figure 3d (i, forward operation). The corresponding probabilities for (010), (011); (100), (101); (110), (111) are ~0.84, ~0.16; ~0.84, ~0.16; ~0.25, ~0.75, indicating that the correct AND logic operations are acquired. However, it can be argued that one of the fundamental assets of any logic operation, i.e., logic correctness, is only probabilistically confirmed. Therefore, it can be questioned what can be the merit of such logic gating using the p-bits? One of the reasonable rationales is the invertible calculation. The following inverted calculation can be performed using the same hardware for AND logic. In this case, y_1 is given first to be 0 or 1, which then requires $x_1, x_2 = (0,0), (0,1), (1,0)$ or (1,1). For this operation, the $V_{IN,y1}$ is fixed to low (5.10 V) or high (5.32 V) voltage, while the $V_{IN,x1}$ and $V_{IN,x2}$ inputs are first settled to 5.23 V and then floated. Then, $V_{IN,x1}$ and $V_{IN,x2}$ values are determined based on their respective input functions, I_{x1} and I_{x2} . Subsequently, similar procedures are repeated to determine the x_1, x_2 for the given y_1 value. The right panel of Figure 3d (ii, inverted operation) reveals that (000), (010), and (100), corresponding to the correct case for $y_1=0$, have a probability of ~0.3, whereas other incorrect cases have a probability of < ~0.1. For the $y_1=1$, the correct and incorrect cases have their respective probability of ~0.5 and < ~0.2. Therefore, it can be inferred that the inverted AND logic operation could be feasibly (statistically)

achieved using the given p-bit circuits. The supplementary information (Supplementary Figs. S3-S17) also shows that inverted logic operations for all the remaining Boolean gates are possible.

Reviewer #2 (Remarks to the Author):

Probabilistic computing, or computing using “p-bits” originally proposed by Datta group from Purdue University has gained some traction as one of the approaches to implement an Ising Machine, which themselves are investigated for solving certain problems that can be cast or recast as a global optimization problem. A p-bit computer is “wired” up corresponding to the optimization program and then the result is obtained by either through simulated annealing or via very long term sampling of the p-bit values which seeks to obtain the bit pattern corresponding to the ground-state (lowest eigen value) of the Hamiltonian being encoded by the p-bit network. As such this form of computing is suitable for any probabilistic algorithm that is based on energy minimization and a nano-material that shows significant stochastic electrical response is well suited to provide the stochastic samples necessary for the operation of such an optimizer.

Datta group has presented quite a few papers on the theoretical analysis, as well as low barrier magnet MTJ based hardware (published in Nature), as well as emulated p-bits generated purely algorithmically using CMOS circuits, including microcontrollers, and FPGA (from Camsari group at UCSB). It should also be noted that p-bits do not provide the only implementation for an Ising machine solver as many other groups have approached this problem using dynamical systems built from oscillators built from CMOS, variety of memristors, optical substrates, charge density waves in 2D materials etc. It should also be noted that a significant number of quantum computing research, most prominently D-Wave, are solving the same problem via quantum methods (quantum annealing rather than simulated annealing) even though the problem itself is classical and small problems can be solved using simple MATLAB codes on a mid-range laptop. Therefore the jury is still out if this method has any intrinsic benefits over conventional computing and at what problem sizes might such an advantage show up.

The authors have presented a work that attempts to build proof-of-concept Ising chains using a Copper-Telluride-Platinum Hafnia (CTHP) diffusive memristor. They have fabricated p-bits using a combination of the CTHP, a resistor, and a comparator. This is in lines of other designs presented using MTJs. The circuits were assembled on a breadboard, which is also as per the standard practice. Therefore I find the methodology section as per the standard setups being used in this area.

→ Answer from authors:

We appreciate this brief introduction to the background of p-computing and summarization of the manuscript. The reviewer has a deeper understanding of this field, which we appreciate very much. The point-by-point responses to each comment are as follows.

Comment 1: However my central concerns are against the presentation of the material, going beyond the stylistic approaches. The authors present the p-bit with a comparison between classical/deterministic, p-computing and quantum computing. This comparison starts right at the very beginning, e.g. line 34-39. While the quantum computing has indeed being used in Ising machine like problem as I mentioned before, this contentions is not warranted. Quantum computing's power arises from large scale entanglement which really forms a large state space to compute over. Similarly quantum computers often exploit "negative" probability amplitudes that allow for destructive interference of the compute possibilities (see Grover search or the classic Deustch algorithm that started off the field). I suspect that authors are motivating the p-bits as a "poor man's quantum computer" since quantum computers are significantly harder to build and operate. **However a much simpler motivation of solving BPP algorithms, going beyond P ones would have been sufficient.** There is also a strong conjecture that $P=BPP$ which, if true, will render such accelerators moot. **Therefore table 1 is not illuminating beyond trivial comparison.**

→ Answer from authors:

We thank the referee for the valuable comments. We organized Table I to enhance the comprehension of p-bit computing, which might be uneasy to understand for most readers unfamiliar with quantum computing. We agree that p-computing is similar to quantum annealing. Both computing schemes adopt the same energy-based model or Hamiltonian to acquire the solution to the target problem. We added an introductory paragraph about quantum annealing and its connection to p-computing as below.

→ Page 6, added and modified:

The operation principle of p-computing is derived from the theoretical background of quantum computing called quantum annealing, which shares similar features with a stochastic Hopfield network.³⁵ Quantum annealing is based on an energy-based model to solve combinatorial optimization problems. In quantum annealing, the energy of a quantum system comprised of "qubits" is defined as Hamiltonians, further divided into the initial and final Hamiltonian. The initial Hamiltonian denotes the initial ground state of the system, where each qubit remains in a state with the quantum superposition

of 0 and 1. As the system undergoes the annealing procedure, the initial Hamiltonian slowly develops into the final Hamiltonian, which provides a low-energy solution to the given problem.³⁶ Similarly, for p-computing, each qubit can be substituted with multiple “p-bits,” in which the binary states fluctuate with time.^{37,38} For simplicity, the equations of a stochastic Hopfield network or a Boltzmann machine³⁹ are introduced, which may provide the conceptual framework of the present p-computing principle.

Comment 2: Another such concerning point is the comparison made between machine learning methods and p-computing. It is puzzling to me because authors themselves correctly point out that p-computers are merely stochastic Hopfield networks, a very popular neural network, also called the Boltzmann machine. **The operating principles of operation is precisely the same**, energy minimization and a Boltzmann distribution based programming of energy eigen values. Even the original proponent of p-bit have clearly stated that their approach to actually figure out the weights depend on the same algorithm that is well known and used in Hopfield network literature, i.e. carefully aligning the eigenvectors and thus the p-bit values to the truth table being programmed. For an unknown problem, some sort of weight update algorithm will be necessary even for p-computing. All this makes me wonder how much thought the authors have given in connecting these established and well known pieces in their own minds, and this reflects in the exposition.

→ Answer from authors:

Thank you for the valuable comments. The way how machine learning and p-computing find the solution to a particular problem is different. In machine learning, the synaptic weights connecting the neurons are updated according to the appropriate training schemes to maximize the probability of getting the correct solution. As a result, after the training is over, the network outputs the solution with high accuracy. In contrast, in p-computing, there is no weight update; once the weights are fixed to the values derived from the energy function, the system finds the p-bit configuration with the highest probability. Each p-bit state fluctuates with time, and the configuration with the highest output frequency is chosen as the solution. We added the following comments to make this point clearer.

→ Page 5, added and modified:

The p-computing can calculate the results in one shot without training the weights. In contrast, machine learning takes multiple training epochs and consequently more power to optimize the weight matrix to calculate the correct results.

We also modified the section **Logic operations** substantially to show the more precise operation of the suggested p-bit circuits, in which the difference was also elaborated. Please refer to our answer to the

last comment of reviewer #1 above.

Comment 3: My last concern is regarding the novelty of the work. The basic principles are well known and established in the literature, only difference is that they have used a particular material to create their p-bits as a drop-in replacement for a BSN which again is a very old and well known concept, see the very first chapter of Simon Haykin's celebrated book on NNs. I am not sure that such an incremental work warrants a publication at a venue such as Nature Communication, given the utter lack of surprise in the presented results which do not illuminate or advance the field in any significant way.

→ Answer from authors:

We admit that what we report in this work may not be regarded as a breakthrough for the new algorithm or computational method for the mentioned complex problems. Instead, we believe we provided a significantly improved implementation method based on the known principle suggested by Datta's group using the 2-terminal diffusive memristors. Compared with the MTJ p-bit in which a transistor-modulated current drives the MTJ device, our diffusive memristor is driven by a direct voltage. Also, we made several breakthroughs compared with the previous works in terms of power, size and fabrication process, also noted in Table II. Another notable merit is that we provided a complete list of cost functions for all the necessary Boolean logic gates, even with a simpler form than the original suggestion by Datta's group, which has not been reported yet. We believe that these are sufficient justification for this work to be seen in this journal, which other reviewers also appreciate.

→ Page 15, modified:

Moreover, the cost functions and the input functions for all 16 Boolean logic operations were derived in a simpler form compared to the previous works. All logic operations were implemented in forward and reversed directions through the memristor-based p-computing network. Complex functions, such as full adder and multiplication/factorization, were also suggested, showing the methodology's potential to be applied to more complex logic circuits. Finally, a comparison between other probabilistic computing hardware and this work is shown in Table II.

Reviewer #3 (Remarks to the Author):

I think this manuscript reports significant new results using memristive elements in a role that is very different from the usual synaptic functions. As such I believe it should be published in a journal like

Nature Communications after appropriate revisions.

→ Answer from authors:

We appreciate the reviewer for the positive evaluation of our work.

Comment 1: There are statements in this paper that do not sound right to me. For example, in the context of quantum computing I see the statement: “.. lacking the general algorithm for Boolean logic operations is another critical huddle for its widespread use.” **I believe that at least some of the probabilistic algorithms demonstrated here (like Boolean operations and factorization) were first explored in the context of adiabatic quantum computing.**

→ Answer from authors:

Thank you for the correction. We deleted the statement from the main text.

→ Page 2, deleted:

~~Also, lacking the general algorithm for Boolean logic 36 operations is another critical huddle for its widespread use.~~

Comment 2: One minor comment is about their use of “Hopfield networks” to describe what is probably a “Boltzmann machine” which is a stochastic Hopfield network.

→ Answer from authors:

We appreciate this careful comment. We agree that the repeated use of “Hopfield Network” may confuse the reviewer and readers. We made corrections and added a few citations regarding the hardware implementations of Boltzmann networks to prevent confusion.

→ Page 5, modified:

The operation principle of p-computing is derived from the theoretical background of quantum computing called quantum annealing, which shares similar features with a stochastic Hopfield network.

→ Page 5, added:

For simplicity, the equations of a stochastic Hopfield network or a Boltzmann machine³⁹ are introduced,

which may provide the conceptual framework of the present p-computing principle.

Comment 3: Since this paper uses memristors to implement a function that has been implemented with magnetic tunnel junctions, it seems to me that a paragraph comparing them may be of interest to readers.

→ Answer from authors:

We thank the reviewer for this fruitful advice. Table I was added to assist the readers who may not be well familiar with quantum and probabilistic computing principles. We agree that a more realistic comparison between our work and previous works would be more appropriate to prove the novelty of the manuscript. Therefore, we organized Table II with quantitative data such as power, fabrication and size, shown below.

→ Page 13, added:

Complex functions, such as full adder and multiplication/factorization, were also suggested showing the methodology's potential to be applied to more complex logic circuits. Finally, a comparison between other probabilistic computing hardware and this work is shown in Table II. The power consumption was calculated by the equation: $P = V_{in}I_{avg}$, where V_{in} is the input voltage, and I_{avg} is the average current flowing through the CTHP device. A typical comparator design was considered for the comparison of the cell area.

Table II: Comparison between various probabilistic computing hardwares. For the MTJ and CTHP devices, only energy for the probabilistic device was considered (comparator energy consumption was excluded). However, the operation speed of the MTJ is higher than CTHP.

	CMOS-based¹²	Magnetic Tunnel Junction-based^{12,49}	This work
Fabrication (Device)	Complex (Transistor Logic)	Complex (Ta/Pt/[Co/Pt] ₇ /Co/Ru/[Co/Pt] ₂ /Co/Ta/CoFeB/MgO/CoFeB/Ta/Ru/Ta)	Simple Metal/Insulator/Metal (Cu _x Te _{1-x} /HfO ₂ /Pt)
Power Consumption	200 μW	10 μW	2.3 μW

CMOS circuits (Number of transistors)	LFSR (1194)	NMOS+Comparator (11)	Comparator (10)
--	----------------	-------------------------	--------------------

REVIEWER COMMENTS

Reviewer #1 (Remarks to the Author):

(0) The authors responded to the reviewer comments well. Remarks regarding quantum computing is now tempered and there are comparisons to MTJ-based p-computing. The synaptic operation with FPGA + DAC is clearly visible with details.

The paper evolved to a much better state. However, I still have some suggestions for revision, mostly to ensure the paper remains scientifically accurate in the long term.

(1) The most important one is the use of "quantum computing". It would be safer and more "accurate" to simply replace this with "quantum annealing". For example, Table I compares p-computing with q-computing and suggests they both have high "parallelity of processing" (a better term is needed since I am not sure what this means). The full distinction of gate-based quantum computing vs p-computing was recently explored in <https://arxiv.org/abs/2007.07379>. In that paper the authors showed that emulating a generic quantum gate with p-bits requires polynomial resources in memory but exponential resources in time, more or less in line with the the theoretical computer science understanding. However, Ref. 37 of the manuscript (by the same authors of the article I mentioned) showed that when it comes to quantum annealing, the similarities are much more direct and there is no apparent need for exponentially more resources for p-computing. As such, p-computing and q-computing should not be called out equally like so, as shown in Table~I and in the paper but a much better comparison could be quantum annealing (if such a comparison is absolutely necessary).

Otherwise, remarks from the authors might draw significant criticism from the quantum computing community if the authors publish the manuscript in this form, in such a visible venue!

(2) Table II is a great start. I understand the authors' instinct to put the best aspects of the memristors vs MTJs ... But it would be good to have a column that shows the present-day speeds of MTJs and the prospects of memristors (and how fast they can be) in the future.

Right now, the authors simply added this sentence to the caption without any references that I mentioned in my previous round report: "However, the operation speed of the MTJ is higher than CTHP."

Yes, it is true that stochasticity might be a better alternative than MTJs for p-computing like the authors suggest but a frank assessment is needed. It is understandable that the first experiments

with memristors do not show ~nanosecond fluctuations but are there prospects for this to happen ?
What other advantages might memristors hold ?

(3) Regarding the statement that's added in this round:

"The p-computing can calculate the results in one shot without training the weights. In contrast, machine

learning takes multiple training epochs and consequently more power to optimize the weight matrix to

calculate the correct results."

My suggestion is to remove this or qualify it somehow. When it comes to small invertible logic operations like Full Adders and AND gates ... Machine Learning algorithms can find these weight matrices quite easily, as well. Training a Boltzmann Machine with 5-visible nodes will easily find the Full Adder weights with 5 p-bits.

(4) Regarding the statement:

"Another advantage of this study is that the memristor-based p- computing enables both forward and inverted operations, allowing for expanding its uses for complex operations, such as integer factorization."

An advantage over what? I thought MTJ-based or any other stochastic p-bit will have the same property of being invertible.

This last sentence in that paragraph may not be needed.

(5) My final comment: I believe the paper shows an intriguing possibility of making p-bits with stochastic diffusive memristors. This is a significant result. Instead of trying to "make the case" for p-computing by contrasting it with q-computing, the authors should try to give a direct comparison with alternatives. I hope the authors will find these comments useful and try to improve the paper another time with this spirit.

Reviewer #2 (Remarks to the Author):

The authors have made changes to the manuscript in a positive direction. I think the exposition of the computing paradigm is much cleaner. This form of computing touches upon many other disciplines, which makes it a hard topic to describe in the space of a paper. However, there are still some issues which need clarification/ better explanation. My comments are as follows:

1. Authors describe this memristor as "fast" in line 182. A little before that it seemed that they used pulses of around 200us, which translates to a frequency of p-value generation of around 5kHz (not considering the relaxation period), which is quite low. My 6 year old laptop running MATLAB can generate a million MCMC samples for a 25-node Ising network in a few hundred milliseconds. Could they specify if they are making the comparison against any other specific material technology?

2. The computing demo part seemed to have been done via simulations using fitted V-V characteristics curves. It is however not clear what the simulation technique was, other than that it was done possibly on python. There is a further claim that device-to-device and cycle-to-cycle variability was considered. I feel this is completely inadequate without further explanation of the simulation methodology. Such a simulation should be performed on some version of SPICE that can account for actual nodal voltages and currents in the memristor along with the driving transistors, because without that the reported power values in table II are meaningless speculations.

3. In continuation of the previous comment, while the variability was considered, it is not clear to me what their impact was on the final calculations.

4. The authors have not discussed physical and logical failure modes specific to the CTHP memristor, without which it is hard to judge the viability and importance of this material in this space. I understand if they have not performed such studies, but if they can point to any other studies or their own thoughts on these point, it will be very useful.

5. I did not understand the point being made during comparison with PiM or LiM architectures (line 265). PiMs use very light weight read-out circuits on a column shared read line sense amp to generate logic operations to avoid intermediate readouts/writeins between memory and processor. I do not see why implementing any of these 16 Boolean operations would be significantly simpler in this methodology as compared to PiM. Further PiMs amortize any "complex" logic circuitry over a

large memory array which makes them practically "free". This claims needs to be either dropped or expanded and clarified further to make technical sense.

Overall, I feel that the exposition has been significantly improved with regards to the computing paradigm, but the details on the memristor itself lacks some clarity.

Reviewer #3 (Remarks to the Author):

The revised manuscript seems much improved.

One comment that the authors may wish to consider:

The novelty of the paper lies in the use of memristors in an unconventional role, namely as p-bits instead of MTJ's or other CMOS-based alternatives. It would be good to stress this point more with quantitative comparisons.

We thank the reviewers for the valuable comments, which helped us reorganize the main concepts and improve the quality of the manuscript. We did our best to respond to each comment and modify the manuscript accordingly. Our point-by-point replies to all the comments are provided as shown below.

REVIEWER COMMENTS

Reviewer #1 (Remarks to the Author):

The authors responded to the reviewer comments well. Remarks regarding quantum computing is now tempered and there are comparisons to MTJ-based p-computing. The synaptic operation with FPGA + DAC is clearly visible with details.

The paper evolved to a much better state. However, I still have some suggestions for revision, mostly to ensure the paper remains scientifically accurate in the long term.

→ Answer from authors:

We do appreciate the thoughtful advice. Especially, explanations of the synaptic operation and the connections with quantum annealing are more clarified in this revision, as follows.

Comment 1: The most important one is the use of "quantum computing". It would be safer and more "accurate" to simply replace this with "quantum annealing". For example, Table I compares p-computing with q-computing and suggests they both have high "parallelity of processing" (a better term is needed since I am not sure what this means). The full distinction of gate-based quantum computing vs p-computing was recently explored in <https://arxiv.org/abs/2007.07379>. In that paper the authors showed that emulating a generic quantum gate with p-bits requires polynomial resources in memory but exponential resources in time, more or less in line with the theoretical computer science understanding. However, Ref. 37 of the manuscript (by the same authors of the article I mentioned) showed that when it comes to quantum annealing, the similarities are much more direct and there is no apparent need for exponentially more resources for p-computing. As such, p-computing and q-computing should not be called out equally like so, as shown in Table~I and in the paper but a much better comparison could be quantum annealing (if such a comparison is absolutely necessary).

Otherwise, remarks from the authors might draw significant criticism from the quantum computing community if the authors publish the manuscript in this form, in such a visible venue!

→ Answer from authors:

We appreciate these careful comments. Among various types of quantum computing, quantum annealing provides the most relevant theoretical background of p-computing, as the reviewer correctly indicated. Therefore, we replaced "quantum computing" with "quantum annealing" throughout the text. Regarding the term "parallelity of processing", our definition was the ability to compute two or more tasks simultaneously. However, we concluded it is premature to compare the parallelity of 3 types of computing directly. Therefore, we deleted the row in Table I about "parallelity of processing".

→ Table I, 6th row removed:

Computation methods	Classical computing	Quantum computing	Probabilistic computing
Data expression	0 or 1 deterministic values	Superposition of 0 and 1; an infinite number of states between 0 and 1	Probabilistic 0 or 1
Hardware implementation	CMOS-based digital logic circuits	Computing system based on electron spin resonance	Oscillating digital outputs based on stochastic devices
Output	Deterministic	Probabilistic	Probabilistic
Power consumption	High	High	Low

Comment 2: Table II is a great start. I understand the authors' instinct to put the best aspects of the memristors vs MTJs ... But it would be good to have a column that shows the present-day speeds of MTJs and the prospects of memristors (and how fast they can be) in the future.

Right now, the authors simply added this sentence to the caption without any references that I mentioned in my previous round report: "However, the operation speed of the MTJ is higher than CTHP."

Yes, it is true that stochasticity might be a better alternative than MTJs for p-computing like the authors suggest but a frank assessment is needed. It is understandable that the first experiments with memristors do not show ~nanosecond fluctuations but are there prospects for this to happen ? What other advantages might memristors hold ?

→ Answer from authors:

Thank you for this careful assessment of the revised text. At this moment, the switching speed of the

memristors, especially ion-migration-based ones, including the present work, can hardly exceed that of the MTJs. However, the memristors still possess the potential to overcome the speed limit. The minimum switching time of memristors was reported as short as tens of picoseconds in literature⁴⁶. They are also advantageous when expanded on a larger scale for complex operations. Furthermore, they generally have a larger tolerance for process variation, such as insulating layer thickness variation, than the extremely tight thickness variation allowance of the MTJ. We added a statement in the main text and an additional row in Table II regarding this point.

→ Page 9, revised:

Still, the bit generation speed of memristor-based p-bits can be further improved through device engineering. For example, substituting the insulator with a higher Cu ion diffusivity can further accelerate the CF formation and dissolution processes.²⁹ The switching time of memristors can be as short as tens of picoseconds, showing the potential for fast and low-power computing.^{18,46} The electrode structure of a memristor is much simpler than the MTJ. They also have a larger tolerance for the thickness variation of the insulating layer compared with the extremely tight allowable thickness variation of the insulating layer in MTJ.

→ Table II, added:

	CMOS-based¹²	Magnetic Tunnel Junction-based^{12,53}	Memristor-based
Fabrication (Device)	Complex (Transistor Logic)	Complex (Ta/Pt/[Co/Pt] ₇ /Co/Ru/[Co/Pt] ₂ /Co/Ta/CoFeB/MgO/CoFeB/Ta/Ru/Ta)	Simple Metal/Insulator/Metal (Cu _x Te _{1-x} /HfO ₂ /Pt)
Power Consumption	200 μW	10 μW	154 nW @ 5.4 V
CMOS circuits (Number of transistors)	LFSR (1194)	NMOS+Comparator (11)	Comparator (10)
Potential switching	100 ps ¹²	200 ps ^{18,52}	85 ps ^{18,46}

speed of the device			
--	--	--

→ Page 15, added:

Based on a 2-terminal metal-insulator-metal structure, p-bits built with memristors are advantageous in area efficiency and production cost compared with other p-bits when integrated into a larger network.

Comment 3: Regarding the statement that's added in this round:

"The p-computing can calculate the results in one shot without training the weights. In contrast, machine learning takes multiple training epochs and consequently more power to optimize the weight matrix to calculate the correct results."

My suggestion is to remove this or qualify it somehow. When it comes to small invertible logic operations like Full Adders and AND gates ... Machine Learning algorithms can find these weight matrices quite easily, as well. Training a Boltzmann Machine with 5-visible nodes will easily find the Full Adder weights with 5 p-bits.

→ Answer from authors:

Thank you for this careful comment. We wanted to emphasize that p-computing is distinguished from machine learning in that it does not require an iterative training step, not directly to compare which one is better. P-computing shares similarities with the Boltzmann machine, but its primary purpose is to solve problems with fixed energy (cost) functions. In contrast, machine learning trains weights to optimize the network to return results with high accuracy. Therefore, power consumption and computing accuracy are in a trade-off relation. We agree that machine learning can perform simple tasks quite easily. To avoid unnecessary complications, we tone down the relevant text as follows.

→ Page 7, revised:

The main difference between p-computing and machine learning is that the p-computing can calculate the results in one shot without training the weights at the expense of the loss of computing accuracy. In contrast, machine learning takes multiple training epochs, which consumes more power. However, machine learning can perform the tasks more accurately by optimizing the weight matrix.

Comment 4: Regarding the statement:

"Another advantage of this study is that the memristor-based p- computing enables both forward and inverted operations, allowing for expanding its uses for complex operations, such as integer

factorization."

An advantage over what? I thought MTJ-based or any other stochastic p-bit will have the same property of being invertible.

This last sentence in that paragraph may not be needed.

→ Answer from authors:

Thank you for the corrections. We intended to mention one of the strengths of memristor-based p-bit, but the context may confuse the readers. Since the inverted operation was already demonstrated in this work and in the MTJ-based work, we deleted this sentence.

→ Page 4, deleted:

~~"Another advantage of this study is that the memristor-based p-computing enables both forward and inverted operations, allowing for expanding its uses for complex operations, such as integer factorization."~~

Comment 5: My final comment: I believe the paper shows an intriguing possibility of making p-bits with stochastic diffusive memristors. This is a significant result. Instead of trying to "make the case" for p-computing by contrasting it with q-computing, the authors should try to give a direct comparison with alternatives. I hope the authors will find these comments useful and try to improve the paper another time with this spirit.

→ Answer from authors:

Thank you for the thoughtful and insightful evaluation of our work. We believe this comment is in line with the 2nd comment above. The memristors are advantageous for complex operations when expanded on a larger scale due to their simple 2-terminal MIM structure. Therefore, it can enhance the area efficiency and reduce production costs. Regarding the device switching speed, the memristors cannot exceed the performance of the MTJs so far. However, memristors still possess the potential to overcome the deficiency. We hope that the modifications in this revision make the reviewer satisfied.

Reviewer #2 (Remarks to the Author):

The authors have made changes to the manuscript in a positive direction. I think the exposition of the computing paradigm is much cleaner. This form of computing touches upon many other disciplines, which makes it a hard topic to describe in the space of a paper. However, there are still some issues which need clarification/ better explanation. My comments are as follows:

→ Answer from authors:

Thank you for the overall positive evaluation of the revised manuscript. The previous comments helped enhance the quality of the manuscript substantially. Below are the responses to each comment and the corresponding corrections in the main text.

Comment 1: Authors describe this memristor as "fast" in line 182. A little before that it seemed that they used pulses of around 200us, which translates to a frequency of p-value generation of around 5kHz (not considering the relaxation period), which is quite low. My 6 year old laptop running MATLAB can generate a million MCMC samples for a 25-node Ising network in a few hundred milliseconds. Could they specify if they are making the comparison against any other specific material technology?

→ Answer from authors:

Thank you for this critical comment. We suppose the intention of the statement was not conveyed smoothly. We intended to imply the potential to enhance the bit generation speed of memristor-based p-bits through additional device engineering. However, it was reported that the switching time of the memristors could be as short as tens of picoseconds⁴⁶. The original explanation may confuse the reviewer, so we changed the text to clarify our intentions.

→ Page 9, revised:

Still, the bit generation speed of memristor-based p-bits can be further improved through device engineering. Substituting the insulator with a higher Cu ion diffusivity can further accelerate the CF formation and dissolution processes.²⁹ The switching time of memristors can be as short as tens of picoseconds, showing the potential for fast and low-power computing.^{18,46}

Comment 2: The computing demo part seemed to have been done via simulations using fitted V-V characteristics curves. It is however not clear what the simulation technique was, other than that it was done possibly on python. There is a further claim that device-to-device and cycle-to-cycle variability was considered. I feel this is completely inadequate without further explanation of the simulation methodology. Such a simulation should be performed on some version of SPICE that can account for

actual nodal voltages and currents in the memristor along with the driving transistors, because without that the reported power values in table II are meaningless speculations.

→ Answer from authors:

Thank you for indicating this critical point. We attempted to conduct an additional HSPICE simulation to calculate the actual power consumption of the circuit. However, it was challenging to achieve the device outputs with the multiple input pulses in HSPICE since the stochastic CTHP switching cannot be modeled by simple circuit elements, such as the resistors and the capacitors, which was necessary to perform the HSPICE simulation. Also, it was challenging to estimate the comparator power since the comparator cannot be realized in the HSPICE simulation environment unless specific transistor characteristics are provided. We consulted this problem with several simulation specialists in our EE department, but they also replied similarly. Because of these problems, we could only estimate the power consumption of the CTHP device and the series resistor by simple mathematical calculations using Ohm's law. The measured pulse data of the CTHP device was averaged over 40 cycles to calculate the circuit's power, as shown in Supplementary Fig. S19. In this case, the power consumptions of the CTHP and series R were added.

→ Page 8, added:

The average power consumption of the p-bit circuit was calculated using the pulse output of the device (Supplementary Fig. S19).

→ Table II, revised:

Table II: Comparison between various probabilistic computing hardware. For the MTJ and CTHP devices, only energy for the probabilistic device was considered (comparator energy consumption was excluded). However, the operation speed of the MTJ is higher than CTHP.

	CMOS-based¹²	Magnetic Tunnel Junction-based^{12,53}	Memristor-based
Fabrication (Device)	Complex (Transistor Logic)	Complex (Ta/Pt/[Co/Pt] ₇ /Co/Ru/[Co/Pt] ₂ /Co/Ta/CoFeB/MgO/CoFeB/Ta/Ru/Ta)	Simple Metal/Insulator/Metal (Cu _x Te _{1-x} /HfO ₂ /Pt)
Power Consumption	200 μW	10 μW	154 nW @ 5.4 V

CMOS circuits (Number of transistors)	LFSR (1194)	NMOS+Comparator (11)	Comparator (10)
Potential switching speed of the device	100 ps ¹²	200 ps ^{18,52}	85 ps ^{18,46}

→ Supplementary Figure S19 added:

Supplementary Figure S19: Power consumption calculation results of the CTHP p-bit circuit.
a) Power consumption of the CTHP device. b) Power consumption of the series resistor. c) The CTHP p-bit circuit design used in the analysis. d) Power consumption at different input voltage pulse amplitudes, averaged over 40 cycles.

Comment 3: In continuation of the previous comment, while the variability was considered, it is not clear to me what their impact was on the final calculations.

→ Answer from authors:

Thank you for this helpful advice. We considered the variability of the device to conduct the experiment assuming more practical situations. Most stochastic elements based on physical phenomena have

intrinsic variations due to the intrinsic and extrinsic noises. These may include variations in device fabrication, thermal noise, parasitic components in the measurement environment, etc. Therefore, we considered these in the first place to ensure the plausibility of the simulation.

Comment 4: The authors have not discussed physical and logical failure modes specific to the CTHP memristor, without which it is hard to judge the viability and importance of this material in this space. I understand if they have not performed such studies, but if they can point to any other studies or their own thoughts on these point, it will be very useful.

→ Answer from authors:

Thank you for your valuable comments. The physical breakdown of the CTHP device can be defined as the permanent change of the switching mode from threshold switching to resistive switching. This behavior is due to the excessive inclusion of Cu atoms into the insulating layer. To ensure stable generation of the p-bit outputs, we measured the endurance, estimated to be over 10^6 cycles (Supplementary Figure S1e). We also noted this aspect in the "Memristor-based p-bit and p-computing system" section. Moreover, the endurance of Cu-based threshold switching can reach 10^{10} cycles, according to literature⁴⁵.

→ Page 8, added:

The endurance of Cu-based threshold switching can reach 10^{10} cycles, showing the potential for stable bit generation.^{44,45} However, its threshold switching performance could be frustrated by changing it to the resistive switching mode, accompanied by the excessive Cu atom migration into the insulating layer.

Comment 5: I did not understand the point being made during comparison with PiM or LiM architectures (line 265). PiMs use very light weight read-out circuits on a column shared read line sense amp to generate logic operations to avoid intermediate readouts/writeins between memory and processor. I do not see why implementing any of these 16 Boolean operations would be significantly simpler in this methodology as compared to PiM. Further PiMs amortize any "complex" logic circuitry over a large memory array which makes them practically "free". This claims needs to be either dropped or expanded and clarified further to make technical sense.

Overall, I feel that the exposition has been significantly improved with regards to the computing paradigm, but the details on the memristor itself lacks some clarity.

→ Answer from authors:

Thank you for pointing out this critical point. We agree that comparing our p-bit with the PiM or LiM is too premature. We deleted the sentence from the main text.

→ Page 13, deleted:

~~Also, the memristor-based LIM requires additional operation steps for a more complex operation, such as a full adder.~~

Reviewer #3 (Remarks to the Author):

The revised manuscript seems much improved.

One comment that the authors may wish to consider:

The novelty of the paper lies in the use of memristors in an unconventional role, namely as p-bits instead of MTJ's or other CMOS-based alternatives. It would be good to stress this point more with quantitative comparisons.

→ Answer from authors:

We thank the reviewer for the favorable review of our research. Since we already compared with other types of p-bit hardware, we added a statement in the main text to emphasize the strength and perspective of memristor-based p-bits. Although the bit generation speed of the CTHP p-bit is measured as a microsecond-scale, there is still much room for improving device switching speed. The switching speed can be further enhanced through device engineering. Furthermore, the maximum switching time of memristors can be as short as the picoseconds scale in the literature, which shows the prospect of the memristors performing computing efficiency comparable to their counterparts.

→ Page 9, revised:

Still, the bit generation speed of memristor-based p-bits can be further improved through device engineering. Substituting the insulator with a higher Cu ion diffusivity can further accelerate the CF formation and dissolution processes.²⁹ The switching time of memristors can be as short as tens of picoseconds, showing the potential for fast and low-power computing.^{18,46} The electrode structure of a memristor is much simpler than the MTJ. They also have a larger tolerance for the thickness variation of the insulating layer compared with the extremely tight thickness control of the insulating layer in MTJ.

REVIEWERS' COMMENTS

Reviewer #1 (Remarks to the Author):

I think the authors have revised the manuscript well, addressing concerns from all the reviewers. I have no further comments, the paper is balanced and reads well.

Reviewer #2 (Remarks to the Author):

The paper has significantly improved and the exposition is much clearer. The analysis of variability and endurance is a significant addition which I feel has made this a useful paper in this space. In lines with the other reviewer's comments, I think the reduction on the novelty and utility claims for the p-computing and focusing more on the CTHP memristor itself as a platform has made this a significantly better read.

I believe that the paper is acceptable and meets the standards of Nature Communications in terms of importance, relevance, and novelty.

As an optional suggestion to the authors, it is indeed possible to model the CTHP and other memristors (and indeed most other trivial and non-trivial dynamical systems) within a SPICE like environment. However it is a non-trivial effort especially if the specific expertise is lacking within the department since it is a highly unconventional area as yet for the electrical engineering and needs an extensive search for appropriate modeling abstractions. One example of such an approach is the "Modular Approach to Spintronics" which was published in a sister journal of NCOMMS a few years ago. This does not distract from the present work and its importance, and my suggestion is for any future work authors may undertake.

I wish my best to the authors.

We appreciate the valuable advice from the reviewers, which significantly enhanced the quality of this work and straightened out the ideas.

REVIEWER COMMENTS

Reviewer #1 (Remarks to the Author):

I think the authors have revised the manuscript well, addressing concerns from all the reviewers. I have no further comments, the paper is balanced and reads well.

→ Answer from authors:

We thank the reviewer for the thoughtful comments, which helped the manuscript, especially in refining the adiabatic quantum computing concepts.

Reviewer #2 (Remarks to the Author):

The paper has significantly improved and the exposition is much clearer. The analysis of variability and endurance is a significant addition which I feel has made this a useful paper in this space. In lines with the other reviewer's comments, I think the reduction on the novelty and utility claims for the p-computing and focusing more on the CTHP memristor itself as a platform has made this a significantly better read.

I believe that the paper is acceptable and meets the standards of Nature Communications in terms of importance, relevance, and novelty.

As an optional suggestion to the authors, it is indeed possible to model the CTHP and other memristors (and indeed most other trivial and non-trivial dynamical systems) within a SPICE like environment. However it is a non-trivial effort especially if the specific expertise is lacking within the department since it is a highly unconventional area as yet for the electrical engineering and needs an extensive search for appropriate modeling abstractions. One example of such an approach is the "Modular Approach to Spintronics" which was published in a sister journal of NCOMMS a few years ago. This does not distract from the present work and its importance, and my suggestion is for any future work authors may undertake.

I wish my best to the authors.

→ Answer from authors:

We appreciate the precious comments throughout the revision process of this work. Also, we thank the reviewer for suggesting the idea for future works. We will consider conducting the advanced simulation method using the HSPICE model of the CTHP device.